# On the Contribution of Fast and Slow Responses to Precipitation Changes Caused by Aerosol Perturbations

Shipeng Zhang[1], Philip Stier[1], Duncan Watson-Parris[1]

[1] Atmospheric, Oceanic and Planetary Physics, Department of Physics, University of Oxford, UK

*Correspondence to*: Shipeng Zhang (shipeng.zhang@physics.ox.ac.uk)

**Abstract.** Changes in global-mean precipitation are strongly constrained by global radiative cooling, while regional rainfall changes are less constrained because energy can be transported. Absorbing and non-absorbing aerosols have different effects on both global-mean and regional precipitation, due to the distinct effects on energetics. This study analyses the precipitation responses to large perturbations in black carbon (BC) and sulphate (SUL) respectively by examining the changes in atmospheric energy budget terms on global and regional scales, in terms of fast (independent of changes in sea surface temperature (SST)) and slow responses (mediated by changes in SST). Changes in atmospheric radiative cooling/heating are further decomposed into contributions from clouds, aerosols, and clear-clean sky (without clouds or aerosols).

Both cases show a decrease in global-mean precipitation, dominated by fast responses in the BC case while slow responses in the SUL case. The geographical patterns are distinct too. The intertropical convergence zone (ITCZ), accompanied with tropical rainfall, shifts northward in the BC case, while southward in the SUL case. For both cases, energy transport terms from the slow response dominates the changes in tropical rainfall, which are associated with the northward (southward) shift of Hadley cell in response to the enhanced southward (northward) cross-equatorial energy flux caused by increased BC (SUL) emission. The extra-tropical precipitation decreases in both cases. For the BC case, fast responses to increased atmospheric radiative heating contribute most to the reduced rainfall, in which absorbing aerosols directly heat the mid-troposphere, stabilise the column, and suppress precipitation. Unlike BC, non-absorbing aerosols decrease surface temperatures through slow processes, cool the whole atmospheric column, and reduce specific humidity, which leads to decreased radiative cooling from the clean-clear sky, and is consistent with the reduced rainfall. Examining the changes in large-scale circulation and local thermodynamics qualitatively explains the responses of precipitation to aerosol perturbations, whereas the energetic perspective provides a method to quantify their contributions.

## 1. Introduction

Aerosols have been proposed to affect clouds and precipitation to a large extent by interacting with clouds and radiation (Ramanathan et al., 2001). However, aerosol effects on clouds and precipitation remain highly uncertain due to the complex nature of aerosol-cloud-radiation interactions. For example, satellite-estimated and model-simulated aerosol-cloud interactions show large discrepancies in terms of magnitude and even in sign (e.g. Ackerman et al., 2004; Rosenfeld et al., 2019; Wang et al., 2012). Disagreements between different studies can be attributed to methodologies (Gryspeerdt et al., 2014), model uncertainties (White et al., 2017) and, importantly, are often related to differences in environmental conditions, such as relative humidity, dynamic background, cloud types, stability (Alizadeh-Choobari, 2018; Khain, 2009; Khain et al., 2008; Lohmann et al., 2007; Zhang et al., 2016). Knowledge about the chain of processes, from aerosol emission to acting as cloud condensation nuclei (CCN) or ice nuclei (IN) and to cloud microphysics and dynamics, is critical for reducing the uncertainties and understanding the climate system (Ghan et al., 2016), which is referred to as a 'bottom-up' approach. However, this is challenging, considering uncertainties can arise from aerosol emissions, activation, cloud microphysics and dynamic regimes (e.g., Gettelman et al., 2013; Ghan et al., 2012; Michibata et al., 2016; Zhang et al., 2016).

An energetic perspective provides an alternative approach to examine aerosol effects on precipitation, which is referred to as a 'top-down' approach. For global scales, in equilibrium, latent heat released from rainfall should be energetically balanced by atmospheric radiative cooling together with surface energy fluxes (Allen and Ingram, 2002; Andrews et al., 2010). Climate forcers, such as greenhouse gases (GHGs) and aerosols, which affect the energy budget, can modify the hydrological responses (Kvalevåg et al., 2013; Stephens and Hu, 2010). The energy constraints can be applied to regional rainfall by introducing the energy transport term (H) (Muller and O'Gorman, 2011; Richardson et al., 2016). The local energy budget at equilibrium can be addressed as the following equation:

$$L\delta P = \delta Q + \delta H \tag{1}$$

where $\delta$ denotes the difference between two climate states (e.g., with and without anthropogenic aerosols). L refers to the latent heat of condensation, and P is the precipitation rate, so LP refers to the atmospheric latent heating rate from rainfall. H is the column-integrated divergence of dry static energy which is expected to be zero on a global scale. Q is the atmospheric diabatic cooling (except for latent heat released from precipitation), consisting of atmospheric radiative cooling (ARC) and downward surface sensible heat flux (-SH). ARC is the difference of shortwave (SW) and longwave (LW) fluxes between top of the atmosphere (TOA) and the surface. ARC has significant impacts on global hydrological sensitivity (Allen and Ingram, 2002), while changes in the energy transport term ($\delta H$) are essential in determining the spatial pattern of precipitation response (Muller and O'Gorman, 2011). Dagan et al., (2019b) further demonstrated that whether precipitation responses are more correlated with changes in Q or H depends on the latitude considered. In the extra-tropics, diabatic cooling/heating perturbations are confined to local scales due to strong Coriolis force (thus weak energy transport), and hence the latent heating must balance diabatic cooling according to the energy budget. However, in the tropics, horizontal gradients of dry static energy are small due to the weak Coriolis force. Therefore, local strong diabatic heating perturbations can lead to thermally direct circulations that drive convergence/divergence of moisture and dry static energy. This low-level convergence of mass and moisture can lead to vertical motion and thus an increase in precipitation. So rainfall does not necessarily have to positively correlate with diabatic cooling (Dagan et al., 2019b).

Absorbing and non-absorbing aerosols can have different effects on each energy budget term, and thus precipitation. On the global scale, black carbon (BC), a strongly absorbing aerosol, can stabilise the atmosphere and suppress precipitation via strong shortwave absorption for short timescales, but also can increase precipitation by warming up the surface temperature on longer timescales (e.g., Pendergrass and Hartmann, 2012). The net effect can be uncertain among GCMs (Samset et al., 2016), and is sensitive to the altitude where the BC are added (Ming et al., 2010). Unlike BC, non-absorbing aerosols, for example sulphate (SUL), reduce precipitation predominantly by decreasing SST on long timescales through the dimming effect, whereby SUL scatters incoming solar radiation back to the space (Boucher et al., 2013; Kasoar et al., 2018). Additionally, the surface sensible heat flux is more sensitive to changes in BC than SUL (Myhre et al., 2018; Richardson et al., 2018). On zonal scales, due to the relatively short lifecycle of aerosols, the radiative forcing caused by aerosols is hemispherically asymmetric, which leads to a warmer northern hemisphere for the BC case and colder one for the SUL case, respectively. As a result, the cross-equatorial energy fluxes lead to the intertropical convergent zone (ITCZ) shifting towards the warmer hemisphere (Wang, 2009; Bischoff and Schneider, 2016; Zhao and Suzuki, 2019; Keshtgar et al., 2020; Zanis et al., 2020). On regional scales, it is also worth noting that SUL is usually more suitable as CCN due to its higher hygroscopicity as compared to BC. It can therefore alter cloud microphysics and subsequent precipitation formationregional rainfall by interacting with clouds. However, the susceptibility of precipitation to sulphate aerosols (and the precursors) shows large discrepancies in satellite-estimated precipitation susceptibility to aerosols from several products (Bai et al., 2018; Haynes et al., 2009), and a broad inter-model spread (uncertainty) in GCMs (Ghan et al., 2016; Samset et al., 2016). Some studies also found that the sensitivity of precipitation to sulphate aerosols variesdiffers between model-simulated and satellite-estimated results, in terms of magnitude and sometimes in sign (Liu et al., 2020; Wang et al., 2012).

These responses of precipitation have been conventionally suggested to be composed of fast and slow responses (Andrews et al., 2009; Bala et al., 2010). Fast responses, on the timescale from days to months, are independent of changes in sea surface temperature (SST), and mostly dependent on instantaneous changes in atmospheric radiative heating/cooling (O'Gorman et al., 2012; Richardson et al., 2016). It should be noted that even though SST is unchanged in atmosphere-only models, the land surface temperature is generally still allowed to vary (Stjern et al., 2017). Slow responses, on the timescale of years, are mediated by changes in sea surface temperature (SST) and strongly correlate with top-of-atmosphere (TOA) forcing (Kvalevåg et al., 2013; Lambert and Webb, 2008; Suzuki et al., 2017). Distinguishing contributions from fast and slow responses are essential for understanding the mechanisms that cause the precipitation changes. For example, Bony et al., (2013) examined the responses of tropical rainfall to increasing GHGs. They found that the fast processes weaken the vertical motion and counteract a considerable part of the increasing trend induced by surface warming. Shaw and Voigt (2015) have investigated predicted changes in the summertime Asian monsoon under a warming scenario caused by GHGs, and the fast responses caused by direct radiative effect are generally opposite to the slow impacts caused by the SST warming. The changes in circulation are essential for local climate responses, including clouds, radiation and precipitation (Johnson et al., 2019), whereas the spatial distribution of aerosols radiative forcing in turn affects atmospheric circulations (Chemke and Dagan, 2018).

Distinguishing contributions from different energetic terms is also helpful for understanding physical processes and model differences (DeAngelis et al., 2015). It has historically been used to distinguish contributions from clouds and aerosols when studying aerosol radiative forcing (Forster et al., 2007; Ghan 2013). While energetics

have been applied before to analyse precipitation responses (e.g., Ming et al., 2010; Dagan et al., 2019b), here we
further decompose them into individual terms to provide additional insights. Changes in the energy transport term
($\delta$H) can be decomposed into eddy and mean state components, which are further associated with changes in
thermodynamics and dynamics (Muller and O'Gorman, 2011; Richardson et al., 2016). Changes in ARC can be
further decomposed into contributions from aerosol (mostly through SW absorption), clouds (LW radiative
cooling), and clear-clean sky (mainly from water vapour, greenhouse gases, and the Planck feedback). While it
has long been appreciated that changes in ARC are essential in balancing latent heat release from precipitation
responses on global scales, their relationship on zonal mean or regional scales (and which ARC component
dominates) has not been fully explored.
The Precipitation Driver Response Model Intercomparison Project (PDRMIP) (Myhre et al., 2017) has conducted
several experiments to study the response of precipitation to different climate forcers, such as GHGs, aerosols,
and solar radiation change (e.g., Samset et al., 2016, Stjern et al., 2018). It has been found that the fast response
dominates the global-averaged precipitation responses to BC perturbation, which differs from other drivers of
climate change (Samset et al., 2017; Stjern et al., 2017). It has also been shown that BC contributes to the most
substantial uncertainties among GCMs in simulating the changes in surface temperature and precipitation, due to
different parameterisations of physical, chemical, and dynamical processes involved on the path from BC
emission to the final climate impact (e.g., Stjern et al., 2017). However, it is worth noting that most PDRMIP
research focuses on global mean changes and addressing uncertainties among GCMs (e.g., Myhre et al., 2017;
Richardson et al., 2018; Stjern et al., 2018). Samset et al., 2016 showed the spatial patterns of fast, slow and total
responses of precipitation to different climate forcers including absorbing and non-absorbing aerosols, with a
greater focus on the inter-comparison between different GCMs and different climate forcers. Here we study the
fast and slow response contribution to total response of precipitation with a focus on the comparison between
absorbing and non-absorbing aerosols, and in particular on the underlying mechanisms causing the differences by
distinguishing contributions from each energetic term at various scales.
In light of previous work illustrating the potential of energy budget constraints for understanding regional
precipitation changes, and the fact that absorbing and non-absorbing aerosols impact the response on two distinct
timescales, we aim to answer three questions: 1. What are the contributions of fast and slow responses to total
precipitation changes on global and regional scales? 2. What is the dominant energetic term in precipitation
responses to absorbing/non-absorbing aerosol perturbations on different spatial and temporal scales? 3. How to
relate changes in local thermodynamics and large-scale circulation to changes in energetic terms and quantify
their contribution to precipitation responses?
**2. Method**
The global aerosol-climate model ECHAM6-HAM2 (Stier et al., 2005, Zhang et al., 2012, Tegen et al., 2019,
Neubauer et al., 2019) is used to perform all the experiments. It is based on the general circulation model
ECHAM6 (Stevens et al., 2013) and is coupled to the aerosol module HAM2 (Stier et al., 2005; Zhang et al.,
2012). A two-moment cloud microphysics schemes is used to prognostically predict the number and mass mixing
ratios for both cloud water and ice (Lohmann et al., 2007; Lohmann and Hoose, 2009). The parametrisations for
convection, including cumulus convection and deep convections, are based on the scheme by Tiedtke (1989) and
Nordeng (1994). The activation of CCN to cloud droplets is adopted from Abdul-Razzak and Ghan (2000), which
is based on Köhler theory (Köhler, 1936). It should be noted that freshly emitted BC is assumed hydrophobic and
does not act as cloud condensation nuclei. However, subsequent condensation of sulfuric acid and mixing with
hydrophilic sulphate aerosols will increase its hygroscopicity so that internally mixed BC particles can activate as
CCN (Stier et al., 2006). In HAM2.3, BC can act as ice nuclei through heterogeneous freezing, but only in the
accumulation and coarse mode (Neubauer et al., 2019). The parameterisation for autoconversion is from
Khairoutdinov and Kogan (2000). There are 16 spectral shortwave bands in the solar radiation scheme, and 14
spectrum bands in the longwave radiation scheme (Pincus and Stevens, 2013). The general circulation model
ECHAM6 provides essential meteorological backgrounds such as temperature, pressure, wind and humidity,
which is coupled to HAM2 for the parameterisations of several aerosol processes such as aerosol activation and
deposition.
Emissions of anthropogenic BC, organic carbon and sulphate are from the Atmospheric Chemistry and Climate
Model Intercomparison Project (ACCMIP) emission dataset (Lamarque et al., 2010), including emissions from
industry, agriculture, aircraft, domestic, ships, and waste. Biomass burning emissions are also from ACCMIP
dataset, including both natural and anthropogenic biomass burning (Lamarque et al., 2010). Dimethyl sulphide
(DMS) emission is interactively related to the 10-meter wind speed and concentration in seawater.  Biogenic
volatile organic carbon, and volcanic emissions are following the AeroCom phase II emission dataset (Dentener
et al., 2006). All the emissions are prescribed for the year 2000, so there are no interannual variabilities of
emissions. Simulations are performed at T63 (1.9° ×1.9°) spectral resolution using 47 vertical levels (L47).
To study the precipitation response to absorbing and non-absorbing aerosol perturbations, we analyse two
scenarios: one with a ten-times increase in BC emissions and another with a five-times increase in sulphur dioxide,
relative to baseline emissions in the year 2000 (Tegen et al., 2019). It should also be noted that the increases of
BC emissions here include both anthropogenic and natural sources. This is because the biomass burning emission,
as a large source of BC, includes both anthropogenic and emissions (e.g. agricultural waste burning) and naturally
occurring wild fire emissions. The anthropogenic contribution to wildfire emissions is assumed to dominate but
subject of significant uncertainties (e.g. Lamarque et al., 2010; van Marle et al., 2017). and it is very uncertain to
separate anthropogenic contribution of wild-fires. However, the increases in SO2 emissions are all anthropogenic
because the sources of volcanic and sulphur are kept the same. The main purpose of this work is to better
understand the mechanisms of aerosol-precipitation interactions, with a focus on, but not limited to, anthropogenic
aerosol effects. As only particular aerosol emissions are changed in each perturbation, the differences between
baseline and the perturbed case can be interpreted as aerosol effects. Geographical patterns of emission aerosol
optical depth change can be found in the supplementary file (Figure S1). We chose the multipliers of aerosol
emissions differently here is to make the aerosol effects statistically large enough and keep their radiative forcing
at the same magnitude (Myhre et al., 2017). Another reason is to make our results comparable with PDRMIP
work (Samset et al., 2016).
We run the simulations for 100 years with a mixed layer ocean (MLO), which is described as 50 meters in depth
(Dallafior et al., 2016). The ocean heat transport term (also known as the Q flux) is prescribed, which also means
the ocean dynamics are unchanged. Therefore, the changes in SST are caused by local responses to net surface
heat flux, and the responses in ocean circulations are omitted. To obtain the equilibrium state of precipitation
responses to aerosol perturbations, i.e. the total response ($\Delta P_{total}$), we use the last 50 years of the simulations

because at that time the model has reached approximate equilibrium (Samset et al., 2016). We acknowledge that it might take more than 100 years for a slab ocean model to fully equilibrate. Therefore we also performed a Gregory-style regression (Gregory and Webb, 2008) to check the equilibrium for the BC and SUL cases respectively (see supplementary file). For the BC experiment, it is very likely to reach equilibrium is reached approximately after 50 years. For the SUL case, the energy imbalance is significantly reduced and reaches a near-equilibrium after 50 years run as well, but it is suggested that more than 100 years simulation is needed to fully equilibrate. So the total response of surface temperature to 5 times SUL should be even lower (more negative). Considering the purpose of our study is to understand the mechanisms of precipitation responses to aerosols, an exact equilibrium is not critical here and our conclusions still apply to an approximate equilibrium. Another simulation is run for 20 years with fixed sea surface temperatures (fSST) and last ten years are used. The precipitation responses for fSST simulations can be interpreted as the fast response ($\Delta P_{fast}$). The slow response is then calculated as the difference between the total response and the fast response (Myhre et al., 2017; Samset et al., 2016):

$$\Delta P_{slow} = \Delta P_{total} - \Delta P_{fast} \tag{2}$$

The length of integration period is sufficient to derive the fast and total responses because the fast response of precipitation occurs on time scales from days to months and a slower response on a time scale of years (Myhre et al., 2017).

Since fast and slow responses are examined from an energetic perspective, we focus on how the atmospheric diabatic cooling (Q) and energy transport terms (H) respond to aerosol perturbations in fSST and MLO simulations. H is calculated offline, as a residual by using the energy budget equation. Following previous studies (e.g., Muller and O'Gorman, 2011; Richardson et al., 2016), Q is the combination of atmospheric radiative cooling (ARC) and downward surface sensible flux (-SH), as follows:

$$Q = ARC - SH \tag{3}$$

ARC is defined as net shortwave (SW) and longwave (LW) radiation loss of the atmospheric column, which can be calculated from the difference between the top of atmosphere (TOA) and surface radiative fluxes (downward positive), defined as

$$ARC = (LW_{TOA} + SW_{TOA}) - (LW_{SUR} + SW_{SUR}) \tag{4}$$

Ghan (2013) suggested using additional diagnostics to distinguish aerosol radiative forcing from aerosols, clouds, and surface albedo. This has been widely adopted in current GCMs to better estimate aerosol effects (e.g., Zhang et al., 2016). Following Ghan (2013), we further decompose ARC into contributions from clouds, aerosols and clear-clean sky (without aerosols and clouds) separately (Equation 5), by using the same additional radiation call to calculate ARC from the clear-clean sky ($ARC_{clear,clean}$):

$$ARC = ARC_{aerosol} + ARC_{cloud} + ARC_{clear,clean} \tag{5}$$

$$ARC_{aerosol} = ARC - ARC_{clean} \tag{6}$$

$$ARC_{cloud} = ARC_{clean} - ARC_{clear,clean} \tag{7}$$

Since ARC consists of radiative heating/cooling from aerosols (mainly through aerosol direct SW absorption), clouds (primarily through cloud LW absorption/cooling), and clear-clean sky (mainly though LW radiative absorption/cooling from GHGs, water vapour, and Planck feedback), it is helpful to systematically study the effect of absorbing and non-absorbing aerosols on each decomposed energy term, and to further connect those to changes in precipitation.

It is worth noting that $\Delta ARC_{aerosol}$ only includes direct interactions with radiation here and is much more sensitive
to absorbing aerosol burden rather than non-absorbing aerosols. Despite the significant negative radiative forcing
at TOA (Boucher et al., 2013), non-absorbing aerosols do not significantly modify atmospheric radiative
absorption, as they act to decrease net SW radiative fluxes at both the surface and TOA in the same way. Non-
absorbing aerosols can affect atmospheric radiative absorption via changing absorbing aerosol life cycles (Stier
et al., 2006), but the impacts can be very small. It should also be noted here that changes in $ARC_{cloud}$ include
aerosol indirect effects (interactions with clouds) on ARC and cloud feedbacks in slow responses, but most of the
changes are from LW radiation from clouds (e.g., Lubin and Vogelman, 2006) rather than SW radiation. And its
magnitude depends on the temperature (height) at both cloud top and bottom as well as on the ice concentration
at cloud top (see Figure S2 for baseline $ARC_{cloud}$). As aerosol effects on convective clouds are not explicitly
simulated in ECHAM6-HAM2 (or most GCMs) yet, changes of $ARC_{cloud}$ from convective clouds are mostly
caused by aerosol-induced changes in dynamics. Baseline $\Delta ARC_{aerosol}$, $\Delta ARC_{cloud}$, and $\Delta ARC_{clear,clean}$ can be
seen in supplementary file (Figure S2, S3, S4).
**3. Results**
**3.1. Global mean responses**
Table 1 shows the global-mean fast, slow, and total responses of the energy budget terms, including atmospheric
latent heat release from precipitation ($L\Delta P$) and other atmospheric diabatic cooling terms, in response to increased
BC and SUL emission for the fSST and MLO simulations, respectively. Globally averaged precipitation is
decreased in both the BC and SUL experiment, and the associated reduced latent heating is primarily balanced by
decreased ARC (Table 1). However, there are some substantial differences between BC and SUL cases after
decomposition into different contributions.
For the BC case, the decreased precipitation from total responses ($L\Delta P$ around -3.26 W m$^{-2}$) is mostly contributed
by fast responses ($L\Delta P$ around -3.64 W m$^{-2}$). Slow responses ($L\Delta P$ around 0.38 W m$^{-2}$) lead to increased but much
smaller in magnitude precipitation changes compared to the fast responses. Previous studies suggest that fast
responses are largely mediated by atmospheric radiative absorption while slow responses scale with surface
temperature change (Samset et al., 2016). An increase of BC emissions can increase atmospheric absorption to a
large extent, which is a near-instantaneous process. This can be seen from the decomposition of ARC, which
shows that the decreased ARC from fast and total responses is mainly due to the increased SW absorption from
BC aerosols ($\Delta ARC_{aerosol}$) (Table 1). However, the change of global-mean surface temperature in the BC case is
small (around 0.4 K). That is because for an increase of BC emissions, reduction of downward SW radiation
largely counteracts increased downward LW radiation from the warmer atmosphere. As a result, the change of
surface temperature is regionally-dependent and globally small (Stjern et al., 2017) (Figure S2). Large changes in
$\Delta ARC_{aerosol}$ and small changes in global-mean surface temperature lead to a dominating contribution from fast
responses to total global-mean rainfall changes for the BC cases.
For the SUL case, the slow response dominates the total response (Table 1). Since SUL is a non-absorbing aerosol,
which decreases net SW radiative fluxes at both the surface and TOA through scattering solar radiation,
atmospheric absorption changes little. Most of the reduced ARC in the total response is from changes in clear-
clean sky radiative cooling ($\Delta ARC_{clear,clean}$) from slow responses mediated via surface flux changes. As SUL

decreases SW radiation reaching the surface, the global-mean temperature decreases around 2K on a relatively long timescale due to the high capacity of oceans (a slow process). Decreased global-mean temperature further leads to reduced $ARC_{clear,clean}$ from decreased atmospheric column temperature (i.e. Planck feedback) (Zelinka et al., 2020), and decreased water vapour content, which is controlled by the Clausius-Clapyron relationship (Suzuki and Takemura, 2019).

The contribution of changes from SH acts to counteract nearly one-third the decreased ARC in fast and total responses for the BC case, which is much larger than that in the SUL case. This is because the absorbing aerosols heat the atmosphere and decrease the temperature difference between near-surface air and the surface, resulting in reduced upward SH fluxes. So changes in SH are also dominated by the fast response, and mainly act to increase precipitation from an energetic perspective, counteracting the decreasing effect induced by ARC in the BC case (Ming et al., 2010).

## 3.2. Regional responses and their contributions

The geographical patterns of precipitation responses are substantially different between BC and SUL, in both the fast and total responses (Figure 1). The patterns are similar to Samset et al., 2016, in which they showed an ensemble result with a focus on inter-comparison among several models and climate forcers. For the total response, it shows a distinct pattern of an ITCZ shift in response to increased BC and SUL emission. ITCZ tends to shift northward in the BC case while southward in SUL case (Figure 1a and 1b). Since BC warms (SUL cools) the northern hemisphere, there is an enhanced southward (northward) cross-equatorial energy flux in responses to the aerosol perturbation, resulting in ITCZ being shifted towards the warmer hemisphere (Bischoff and Schneider, 2016; Wang, 2009). Changes in tropical rainfall are dominated by changes in the Hadley cell in responses to the enhanced cross-hemispheric energy fluxes. Figure 1e and 1f further show that slow response mainly contributes to the ITCZ shift in both cases. This will be further demonstrated in Section 3.3 and 3.4.

The fast response of precipitation in the BC case (Figure 1c) shows a land-sea contrast pattern in the tropics, in which rainfall increases in central Africa while it decreases in the surrounding tropical ocean. Central Africa is one of the main source regions of BC emission through biomass burning, and tenfold increase of BC emissions makes the burden changes significant (Figure S1). The pattern of the fast precipitation response in the BC case is similar to the pattern of rapid precipitation response to $CO_2$ shown in Richardson et al., (2016). But the mechanism is not exactly the same. In the $CO_2$ case, even though SST remains unchanged, $CO_2$ can increase land surface temperature and the land-sea temperature contrast (warmer land and unchanged ocean) leads to a shift of convection to over land (Richardson et al., 2016). For an increase of BC emissions, increased downward LW radiation from the warmer atmosphere is largely counteracted by a reduction of downward SW radiation. As a result, surface temperature is decreased in central Africa (Figure S2), which differs from the $CO_2$ case. But increased BC emission can still warm up the lower troposphere and lead to more ascending motions over Central Africa (Figure S3) (Dagan et al., 2019b; Roeckner et al., 2006). As for the SUL case, the rapid precipitation response shows an opposite land-sea contrast pattern in the tropics, because SUL cools the land temperature (Figure 1d) as land surface temperature is not constrained in fSST runs. However, considering SUL does not directly affect the diabatic heating/cooling in the atmosphere, which differs from BC, the changes are small and not statistically significant over most regions. There are still some exceptions. For example, southeast Asia, which has the largest contribution to SUL emission, and SUL impacts on rainfall through cooling of land temperature as

well as interactions with monsoon (e.g., Wang et al., 2019). Decreased surface temperature over continents, such
as South America, leads to a decrease of precipitation in most land regions as well as an increase in surrounding
oceans (i.e. southeast Pacific Ocean). (Figure 1d).
In the zonal-mean, precipitation is decreased over northern hemispheric mid-latitudes in both BC and SUL cases
for total responses, but different processes contribute to the total response. Most of the precipitation changes over
high latitudes are contributed by fast responses in the BC case (Figure 1g) and slow responses in the SUL case
(Figure 1h). Dagan et al., (2019b) showed different responses of rainfall to aerosol perturbation in the tropics and
extra-tropics. They demonstrated that precipitation responses are more correlated with the energy transport term
(H) in the tropics where heating anomalies can be compensated for by large-scale thermally-driven circulations,
whereas extra-tropical rainfall responses are constrained by radiative cooling in the extra-tropics due to the
stronger Coriolis force (thus weak energy transport). The different contribution from fast and slow processes
between the BC and SUL case indicates different responses in the diabatic cooling in the extra-tropics, and this
will be addressed in Figure 3 and Figure 4 from an energetic perspective.
Figure 2 quantifies how fast and slow responses contribute to total responses of precipitation on regional scales.
We used the response ratio which has also been used in Samset et al., (2016), as follows
$R_{resp} = (|\Delta P_{fast}| - |\Delta P_{slow}|)/(|\Delta P_{fast}| + |\Delta P_{slow}|)$             (8)
If $R_{resp}$ is larger than 0 and close to 1, it means most of the total responses are contributed by fast responses. If
$R_{resp}$ is less than 0 and close to -1, it means slow responses dominates over fast responses. Samset et al., (2016)
showed continental-based results of $R_{resp}$ for different climate forcers, and found the variabilities among models.
Here Figure 2 focuses only on BC and SUL perturbations, and quantitatively gives us the geographical patterns
of contributions from fast and slow responses to total precipitation change. For the BC case, generally the response
over northern hemispheric midlatitudes is consistent with the globally averaged result shown in Table 1, in which
shows that the precipitation change is dominated by fast responses (Figure 2a). It can be seen from Figure 2a that
significant contribution from fast response over North America, northern Atlantic Ocean, Europe, most regions
in China, and north-eastern Pacifica Ocean. However, as for the changes in tropical rainfall, which is associated
with ITCZ shift seen in the total response, slow responses mainly contribute to the northward shift of ITCZ rather
than fast responses in the BC case. One exception is the Central Africa, where the precipitation changes are still
dominated by fast responses, and this will be further examined later. For the SUL case, it has been shown that
total responses are dominated by slow responses, both globally and regionally (Figure 2b). Some exceptions are
some land regions such as America, China and Sahel regions, where the precipitation change is mostly not
significant in total responses.
**3.3. Changes in energy budget terms**
To explain the different mechanisms between BC and SUL in terms of the contribution from fast and slow
responses in more detail, we examine the changes in each energy budget term from Equation 1.
For the BC case, in fast responses, most decreases in Q are located over the main BC source regions such as
Central Africa, Northeast China (Figure 3a and Figure S1). For zonal mean results, after decomposing δQ into
different terms based on Equation 3 and 5, it shows aerosol SW absorption is the major contributor to changes in
Q (Figure 5a). Since BC is a strongly absorbing aerosol, and the effect is near-instantaneous, the changes of Q
lead to decreased precipitation on global and zonal-mean scales and happen through fast responses (Table 1 and
Figure 5a). The zonal mean plot (Figure 3e) shows that fast responses of δQ caused by aerosol absorption (Figure
5a) leads to reduced rainfall, especially over northern hemispheric midlatitudes (red solid line in Figure 3e).
However, on regional scales, the energy transport term acts to play an important role. The geographical pattern of
fast precipitation changes (Figure 1c) is more similar to fast response of δH (Figure 3c) (spatial correlation ~0.9)
than δQ (spatial correlation ~-0.5). The spatial pattern of fast δH (Figure 3c) also shows a land sea contrast in the
tropics as in the precipitation change distribution (Figure 1c), and this is most prominent in Central Africa and
middle Atlantic Ocean. There is a significant increase of rainfall over Central Africa and decrease over the middle
Atlantic Ocean (Figure 1a). This is mostly contributed by fast responses (Figure 1c and Figure 2a). As mentioned,
this pattern is similar to the case of $CO_2$ shown in Richardson et al., (2016). Although BC decreased surface
temperature in Central Africa through fast responses (Figure S2), BC can still warm up the lower troposphere at
central Africa, which results in a thermal driven circulation which favours more convections there. This is
evidenced by Figure 3c which shows the dry static energy flux flow from Central Africa to the middle Atlantic
Ocean (Figure 3c). Dagan et al., (2019b) performed an idealised experiment by adding an absorbing plume in the
tropics, and found a very similar standing wave pattern of precipitation as a response. Examining δH shows that
this is caused by a thermal driven circulation, which favours more convections over central Africa. Positive δH is
consistent with more ascending motions at central Africa (Figure S3). BC warms up the lower troposphere at
central Africa, which results in more ascending motions (Figure S3), and the dry static energy flux flow from
Central Africa to the middle Atlantic Ocean (Figure 3c).
The slow response of δQ leads to a global increase of precipitation (Figure 3b), but the magnitude is an order of
magnitude less than the fast response in δQ. This increased precipitation in the slow-response is caused by the
associated increase global temperature (Figure 6c) (Table 1). From an energetic perspective, it is mainly associated
with the clear-clean sky LW cooling (ARC$_{clear,clean}$) (Table 1 and Figure 5b) as a result of increased atmospheric
column temperature (Planck feedback). As precipitation responses in the extra-tropics are more correlated with
δQ , larger fast responses of Q explain why rainfall responses in extra-tropics are dominated by the fast response
in the BC case (Figure 2a). Figure 3e shows that the ITCZ shift seen in total responses is strongly correlated with
slow responses of δH. Warmer northern hemisphere caused by an increase in BC leads to a southward cross-
equatorial energy flux, which is accompanied by a northward shift of Hadley cell (Bischoff and Schneider, 2016).
Changes in vertical pressure velocity can be found in Figure 6, which also indicates a northward shift of the
ascending branch of the Hadley cell. From an energetic view, the changes in vertical pressure velocity drive the
dynamic effect on advection of dry static energy, which is a strong component in the changes of divergence of
dry static energy fluxes (δH) in the tropics (Richardson et al., 2016).
For the SUL case, most of the fast responses are not statistically significant (Figure 4a and 4c), and total responses
are dominated by the slow response. For changes in extra-tropics, changes in Q are correlated with changes in
precipitation. SUL decreases the mean-state temperature of troposphere through slow responses, which leads to a
reduction of specific humidity (Figure 7). From an energetic view, it leads to a decreased clean-clear sky radiative
cooling (ARC$_{clear,clean}$) (Figure 5d), which contributes to most of the reduced slow responses of δQ. For changes
in the tropics, like the BC case, slow responses of δH are consistent with the southward ITCZ shift in the total
response (Figure 4d). In the extra-tropics, for the SUL case, there is also an interesting land-sea contrast in both
fast and slow δH, with dry static energy fluxes generally diverging from oceans to lands in fast δH (Figure 4c)
and converging in slow δH (Figure 4d). This is because in the fixed SST simulations, land surface temperature is
still allowed to decrease in response to increased SUL emission (Figure S5b) as a result of reduced downward
SW radiation. The land-sea contrast of temperature (colder land) results in more downward large-scale motions
and divergence of moisture (See Figure S6 for changes in vertical pressure velocity and column-integrated water
vapour) over most land regions, particular Southeast Asia and South America, in fast responses. Since fast
responses have already accounted part of land temperature reduction, ocean surface temperature decreases more
than land surface in slow responses (Figure S2d). The colder ocean temperature therefore leads to an opposite
land-sea pattern compared to fast responses (Figure 4d).
Changes of Q are more robust in the fast response for the BC case, and the slow response of Q is more robust for
the SUL case. Decomposition of diabatic cooling shows its global-mean decrease is dominated by an increase of
atmospheric aerosol absorption for fast responses in BC case (Figure 5a) and decreased radiative cooling from the
clear-clean sky for slow responses in the SUL case (Figure 5d). The decreased $ARC_{clear,clean}$ are mainly caused
by the decreased atmospheric column temperature (Planck feedback) and associated reduced water vapour content
(controlled by the Clausius-Clapyron relationship). Sensible heat flux (upward) is also reduced due to the warmer
atmosphere caused by absorption from BC (Figure 5a).
It should also be noted that changes in diabatic cooling counteract the latent heat released from precipitation
associated with the ITCZ shift in both cases (Figure 3b and Figure 4b). This is mainly caused by $ARC_{clouds}$, as it
contributes a large part of diabatic cooling over tropical regions (Figure 5b and 5d). This counteraction with the
ITCZ shift is caused by the associated change of deep convective clouds (see supplementary file for changes in
cloud properties). This is consistent with the results shown in Naegele and Randall, (2019). They found a negative
correlation between tropical rainfall and diabatic cooling and demonstrated this is caused by feedbacks from deep
convective clouds. More high clouds lead to a decrease of atmospheric LW radiative cooling but an increase of
precipitation, and the negative correlation is robust over tropical regions where deep convective clouds prevail
(Naegele and Randall, 2019). The spatial patterns of fast, slow and total responses to $\Delta ARC_{aerosol}$, $\Delta ARC_{cloud}$,
and $\Delta ARC_{clear,clean}$ can be found in supplementary file.
**3.4. Responses of large-scale circulation and local thermodynamic conditions**
Figure 3e and Figure 4e show that changes in tropical rainfall are strongly associated with slow responses of the
energy transport term, independent of aerosol types (absorbing or non-absorbing), whereas changes in mid-
latitude precipitation are dependent on aerosol types, which are dominated by fast responses of aerosol SW
absorption in the BC case and slow responses of clear-clean sky radiative cooling in the SUL case. To help
understand the mechanisms of the tropospheric response in different regions, we study the response of the large-
scale circulation and thermodynamic conditions, by examining the changes in vertical pressure velocity ($\omega$),
temperature $T$, and specific humidity $q$ (Figure 6 and Figure 7). The vertical pressure velocity ($\omega$) at 500hPa is a
useful method to distinguish different cloud dynamic regimes, and a metric to quantify the strength of large-scale
circulation (Bony and Dufresne, 2005; Zhang et al., 2016). Here we only show zonal mean analysis.
As shown in Figure 6, BC warms up the atmosphere through SW absorption, and the warming is confined mainly
in the Northern Hemisphere (NH) where the BC emissions prevail. This leads to southward cross-equatorial
energy fluxes and northward shift of the Hadley cell (Wang, 2009; Bischoff and Schneider, 2016; Zhao and Suzuki,
2019). The changes in $\omega$ demonstrate the northward shift of the ascending branch of the Hadley cell, which show
an increased upward motion in NH tropics and decreased ascending motion in SH tropics (Figure 6d). Therefore,

the tropical rainfall associated with ITCZ changes in response to the changes of large circulation. Figure 6f further demonstrates that slow responses contribute to most of the changes in tropical large-scale circulations in Figure 6d. It is consistent with Figure 3 that changes in tropical latent heat released from precipitation is mostly contributed by $\delta H(slow)$, because the dynamic component associated with changes vertical velocity dominates the energy transport term over tropics (Richardson et al., 2016). Outside the tropics, changes in $\omega$ are not as significant as in tropics (Figure 6d), and zonal mean rainfall is more related to local changes in thermodynamic conditions. Absorbing aerosols directly heat the mid-troposphere through fast processes (Figure 6b). Heating the mid-troposphere will stabilise the column and suppress precipitation. This is consistent with the energetic perspective shown in Figure 3 and Figure 5a that fast responses of radiative cooling caused by BC SW absorption (reduced $ARC_{aerosol}$) accounts for the decreased latent heat in extra-tropics. An interesting aspect here is that while BC induces the ITCZ shift, the fast response (Figure 6e) seems to counteract the stronger slow response shown in Figure 6f. This is because of the strong non-zonal effect from Central Africa (see geographical pattern of vertical pressure velocity changes in the supplementary file), where BC warms up the lower troposphere resulting in more ascending motions in fast responses (Figure S6). It is also consistent with Figure 1g that fast responses of rainfall in southern tropical branch act to enhance ITCZ while only northern branch act to decrease ITCZ.

For the SUL case, the tropical rainfall response is opposite to that in the BC case, but the mechanism is similar. Increasing sulphate aerosols induces a dimming effect and causes a negative radiative forcing at the surface, which is as fast process. Subsequently, global surface temperatures are decreased, a slow process controlled by ocean heat capacity, , and this cooling is more significant in NH (Figure 7a and 7c). As a result, the northward cross-equatorial energy fluxes lead to a southward shift of the Hadley cell (Figure 7d). The slow responses of the large-scale circulation (caused by SST temperature difference between hemispheres) contributes most of the shift of Hadley cell (Figure 7e). In the extra-tropics, a decrease of precipitation is also found in response to changes in thermodynamics. However, unlike black carbon, SUL decreases surface temperature through slow processes and leads to a cooling of the whole column in the extra-tropics (Figure 7a and 7c). As a result, the specific humidity shows a large reduction (Figure 7i), which is associated with a reduction of rainfall in the extra-tropics. This is consistent with the energetic perspective shown in Figure 4 and Figure 5d that reduced clean-clear sky radiative cooling ($ARC_{clear,clean}$) accounts for the decreased latent heat in extra-tropics.

It is worth mentioning that Figure 6 and Figure 7, as a bottom-up method, qualitatively show how the changes in large-scale circulation and local thermodynamics affect rainfall in terms of total, fast, and slow responses respectively, whereas the energy budget view (Figure 3, 4, and 5), as a top-down method, is easier to quantify these contributions through energetic terms (e.g., the energy transport term, $ARC_{aerosol}$ and $ARC_{clear,clean}$). Combining these two methods makes the link between precipitation and aerosols explicit.

**4. Conclusions**

We have examined the response of precipitation to absorbing and non-absorbing aerosol perturbations by separately increasing BC emission and SUL emission in ECHAM6-HAM2 by 10-times and 5-times their baseline emission, following the PDRMIP protocol (Myhre et al., 2017; Samset et al., 2016). The precipitation response is separated into fast (mediated by near-instantaneous changes in atmospheric radiative cooling) and slow responses

(mediated by changes in SST) on both global and regional scales. An energetic perspective has been adopted to
study precipitation changes. Global-averaged energetics have previously been used to study precipitation
responses (e.g., Ming et al., 2010; some PDRMIP work); here, we further decompose atmospheric heating rates
into individual terms separately for fast and slow responses. Changes in atmospheric latent heat release from
precipitation is balanced by changes in atmospheric radiative cooling (ARC), surface sensible heat flux and local
energy transport. We introduce a method, based on Ghan (2013), to further decompose ARC into contributions
from aerosols (through aerosol direct SW absorption), clouds (through cloud LW absorption/cooling), and clear-
clean sky (without aerosols or clouds; mainly though LW radiative absorption/cooling from GHGs, water vapour,
i.e. Planck feedback).
While it has long been appreciated that changes in ARC are essential in balancing latent heat released from
precipitation on global scales, their relationship on zonal mean or regional scales has not been fully explored. For
global means, although SUL and BC have a different sign of radiative forcing at TOA (Boucher et al., 2013), we
found that precipitation is decreased for both cases, which is energetically balanced by reduced atmospheric
diabatic cooling $\delta Q$ (Table 1). This response occurs at different timescales, dominated by fast responses for BC
and by slow responses for SUL. For BC, on the global scale, the most significant effect is that absorbing aerosols
directly heat the mid-troposphere, stabilise the column, and suppress precipitation. Therefore, most of the changes
are due to aerosol absorption ($ARC_{aerosol}$) from fast responses. Meanwhile BC warms up the lower troposphere
and decrease the temperature differences between the surface and near-surface temperature, which results in a
decreased upward sensible heat. Investigating the energy balance, we found this decreased upward surface heat
fluxes from fSST experiment acts to cancel almost one third the decreasing effect caused by increased aerosol
SW absorption. For SUL, although non-absorbing aerosol does not directly affect ARC through aerosol absorption,
the net negative radiative forcing at TOA in fSST experiments and associated surface forcing leads to a decrease
of global surface temperature through slow responses. As a result, it cools the whole atmospheric column,
accompanied by reduced specific humidity, which leads to reduce precipitation. This can also be seen from the
decreased radiative cooling from the clean-clear sky $ARC_{clear,clean}$ in slow responses.
Zonally averaged patterns of precipitation changes for the BC and SUL cases are different (Figure 1). Tropical
rainfall is primarily associated with ITCZ, which shifts northward for BC, and southward for SUL. Extra-tropical
rainfall is reduced in both cases. For BC, slow responses account for most of the changes in tropical rainfall, while
fast responses dominate changes in other regions (Figure 2a). BC warms the northern hemisphere through slow
responses, which leads to a southward energy flux (Bischoff and Schneider, 2016; Rotstayn and Lohmann, 2002).
From an energetic perspective, in the tropics where intense convections and large-scale thermally driven
circulations prevail, slow responses of the energy transport term dominate the changes in tropical rainfall (Figure
3e), which is associated with the northward shift of Hadley cells (Figure 6). Outside the tropics, BC warms up the
mid-troposphere, stabilises the atmosphere (Figure 6) and suppresses precipitation, which is a fast response.
Energetically, different from the tropics, BC induced increased diabatic heating is locally confined due to stronger
Coriolis force. This geostrophic confinement of the diabatic heating associated with increased aerosols shortwave
absorption has to be balanced by reduced latent heat from precipitation (a fast response) (Figure 5a). For the SUL
case, the slow response dominates in nearly all regions (Figure 2b), which is not surprising given that sulphate
aerosol does not directly affect the column diabatic cooling. In the extra-tropics, SUL decreases surface
temperatures, primarily through slow processes, cools the whole column, and reduces specific humidity (Figure

7). From an energetic perspective, this can also be seen from the decreased radiative cooling from the clean-clear sky (without clouds and aerosols) (Figure 5d) due to the reduced water vapour content and decreased atmospheric column temperature (Planck feedback).

There exist some interesting regions where the responses are distinct from globally or zonally averaged results. Rainfall is significantly increased over the Central Africa, in the BC case, together with reduced precipitation over the middle Atlantic Ocean, and this pattern is most prominent in fast responses. This pattern shows clear similarities with the standing wave pattern response of precipitation to an idealised plume of absorbing aerosols in the tropics (Dagan et al., 2019b). Examining δH shows that this is caused by a thermally driven circulation, which favours more convections over central Africa. BC warms up the lower troposphere at central Africa, which results in more ascending motions (Figure S3). The low latitude (thus weak Coriolis force) allows for the dry static energy to be efficiently diverged from Central Africa to the middle Atlantic Ocean (Figure 3c). In the SUL case, while most regions are dominated by slow responses, in some regions, such as most parts of China and South America, rainfall changes are still dominated by fast responses (Figure 2b), where the surface temperature is significantly decreased (Figure S2). This is due to the dimming effect from SUL and associated surface flux changes, and because changes of land surface temperature are not constrained in fSST experiments. Reduced surface fluxes and temperatures therefore lead to a decrease of precipitation over most land regions as well as an increase at surrounding oceans (e.g., southeast Pacific Ocean).

Changes in zonally averaged vertical pressure velocity, temperature profile, and specific humidity (Figure 6 and Figure 7) show consistency with zonally averaged energetics. Changes in vertical pressure velocity indicate a northward shift of the ascending branch of the Hadley cell in the BC case and SUL case. It is consistent with the changes in the divergence of dry static energy fluxes, which is dominated by the changes in vertical velocity (the dynamic component) in the tropics (Richardson et al., 2016). In the extra-tropics, stabilisation induced by BC through fast response is consistent with increased atmospheric radiative heating from aerosol SW absorption. Reduced specific humidity as well as decreased atmospheric column temperature in the SUL case is consistent with decreased radiative cooling from the clean-clear sky. The changes in large-scale circulations and local thermodynamics qualitatively explains the responses of precipitation, whereas the energetic perspective provides a method to quantify and make their contributions explicit.

In summary, we examined the relationship between aerosol-induced changes in atmospheric energetics and precipitation changes across different scales. Generally, changes in ARC and latent heat from precipitation are largely balanced on global and extra-tropics (Dagan et al., 2019b). However, these two terms are less balanced in the tropics due to efficient local energy transport. We introduced a new decomposition method, derived from Ghan (2013), to examine aerosol effects on precipitation. For absorbing aerosols, decreased global-mean and extra-tropical precipitation is associated with increased atmospheric aerosol SW absorption from fast responses, while for non-absorbing aerosols, reduced rainfall is more correlated with decreased clear-clean sky atmospheric radiative cooling from slow responses. This top-down method, together with traditional bottom-up method, can make the link between precipitation and aerosols explicit and quantify contributions to global and regional rainfall changes.

We noted that high resolutions are desirable for the analysis of regional precipitation changes. However, climate models in such configurations have been widely used in this context (the entire CMIP and PDRMIP exercises rely on this) and been shown to have skills in examining regional rainfalls as well as their responses (e.g. Liu et al.,

2018; Myhre et al., 2017; Samset et al., 2016). Increasing resolution while retaining parameterised convection, as
done in many regional climate modelling studies, raises other concerns as many assumptions underlying these
parameterisations are no longer valid (Prein et al., 2015). Ultimately, such work should be conducted in cloud
resolving configurations (which would also allow to couple aerosols directly to the convection, an effect that is
not currently represented) and work is ongoing to develop the required tools. However, it will still be decades
before these are routinely available. In the context of the focus of this work, with focus on constraints from the
energy budget and the underlying physical constraints in general, GCMs are in fact a very robust tool (and
ECHAM6-HAM is, unlike other GCMs or many cloud resolving models, fully energy conserving). We therefore
believe our approach to be robust, in-line with a vast body of literature on this very topic (e.g. Jordan et al., 2018;
Myhre et al., 2017; Roeckner et al., 2006; Samset et al., 2016; Shawki et al., 2018; Samset et al., 2016). We also
note that internal variability on regional scales is significant, in particular in coupled simulations. However, since
we are examining the average of last fifty years results instead of the transient evolution, the impacts from internal
variability should be small in this case. Therefore, this does not take away from our analysis of physical constraints
on precipitation changes.
This metric provides further insights into the model variability in simulating rainfall and their responses to
different climate forcers, as shown by some PDRMIP research (e.g., Richardson et al., 2018; Stjern et al., 2018).
For example, it has been demonstrated that the response from BC perturbation contributes to a large part of the
substantial uncertainties among GCMs in simulating the changes in surface temperature and therefore
precipitation (Stjern et al., 2017). Distinguishing contributions from individual energetic terms is helpful to assess
uncertainties from aerosol absorption, or feedbacks from clouds, water vapour and surface sensible heat flux. This
will improve our understanding of GCMs and the climate system, which will be the focus of our follow-up work.
There exist some caveats when considering real-world implications of our results. The aerosol perturbation
follows the PDRMIP protocol designed to reveal the fundamental mechanisms and to make the aerosol effect
strong enough to be distinguishable from natural variability. However, these perturbations are too large to be
representative for real-world situations, in particular considering anthropogenic $SO_2$ (the precursor of SUL)
emissions that are starting to decrease in South-east Asia (Zheng et al., 2018). As for Northern Hemispheric
midlatitudes, where the population is concentrated, results here show that increased BC or SUL will lead to
decreased precipitation, but this happens at different time scales. Increased BC may lead to a near-instantaneous
decreased precipitation over China or America, while increased SUL will reduce precipitation via the slow
response, modulated by SSTs, at a much longer time scale. In the real world, it should be mentioned that the
anthropogenic emissions create a mixture of absorbing and non-absorbing aerosols, so the changes in rainfall
strongly depend on the time scale and the real-world emission scenario. It should also be noted that the total
responses of precipitation in this work are derived from mixed-layer ocean experiments and therefore differ from
real-world changes involving changes in the ocean circulation. There are several studies that have addressed the
importance of using ocean-coupled models to accurately simulates regional and global precipitation responses
(e.g., Wang et al., 2017; Zhao and Suzuki, 2019).

**Data availability:** The datasets of original simulations are from the ARCHER facility upon request. The data
used to present in this paper can be found at: http://dx.doi.org/10.17632/8n2vj578r2.1 (Zhang, 2021)

**Author Contributions:** SZ carried out the simulations and analyses. DW and PS assisted with the simulations.
SZ prepared the paper with contributions from all co-authors.
**Acknowledgements:** The simulations were performed on the ARCHER UK National Supercomputing Service.
This research was supported by the European Research Council (ERC) project constRaining the EffeCts of
Aerosols on Precipitation (RECAP) under the European Union's Horizon 2020 research and innovation
programme with grant agreement no. 724602. PS also acknowledges funding from the FORCeS project under the
European Union's Horizon 2020 research program with grant agreement 821205. DWP and PS also receive
funding from the European Union's Horizon 2020 research and innovation programme iMIRACLI under Marie
Skłodowska-Curie grant agreement No 860100. DWP also gratefully acknowledges funding from the NERC
ACRUISE project NE/S005390/1. Many thanks to Guy Dagan, Andrew Williams, Duo Chan, and Xianglin Dai
for helpful discussions.

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

**Table 1. ECHAM6-HAM2 simulated multi-annual global averaged fast, slow, and total responses of atmospheric energy budget terms (LP – the atmospheric latent heating rate from rainfall, ARC – atmospheric radiative cooling, SH – sensible heat flux) and surface temperature (T) in response to increase of 10 times black carbon (BC) emission and 5 times sulphate (SUL) emission. ARC has been further decomposed into the contribution from aerosols, clouds and clear-clean sky. All of terms are shown in equivalent precipitation units of mm $d^{-1}$.**


| (mm $d^{-1}$) | L$\Delta$P | $\Delta$ARC | $\Delta$ARC$_{aerosol}$ | $\Delta$ARC$_{cloud}$ | $\Delta$ARC$_{cc}$ | -$\Delta$SH | $\Delta$T (K) |
|---|---|---|---|---|---|---|---|
| fast, 10BC | -0.13 | -0.21 | -0.29 | 0.03 | 0.05 | 0.08 | -0.03 |
| slow, 10BC | 0.01 | 0.02 | -0.01 | 0.00 | 0.04 | -0.01 | 0.39 |
| total, 10BC | -0.11 | -0.18 | -0.30 | 0.03 | 0.09 | 0.07 | 0.35 |
| fast, 5SUL | -0.01 | -0.01 | 0.01 | 0.00 | -0.02 | 0.00 | -0.14 |
| slow, 5SUL | -0.14 | -0.13 | 0.00 | 0.02 | -0.15 | -0.01 | -1.73 |
| total, 5SUL | -0.15 | -0.14 | 0.01 | 0.01 | -0.17 | -0.01 | -1.87 |






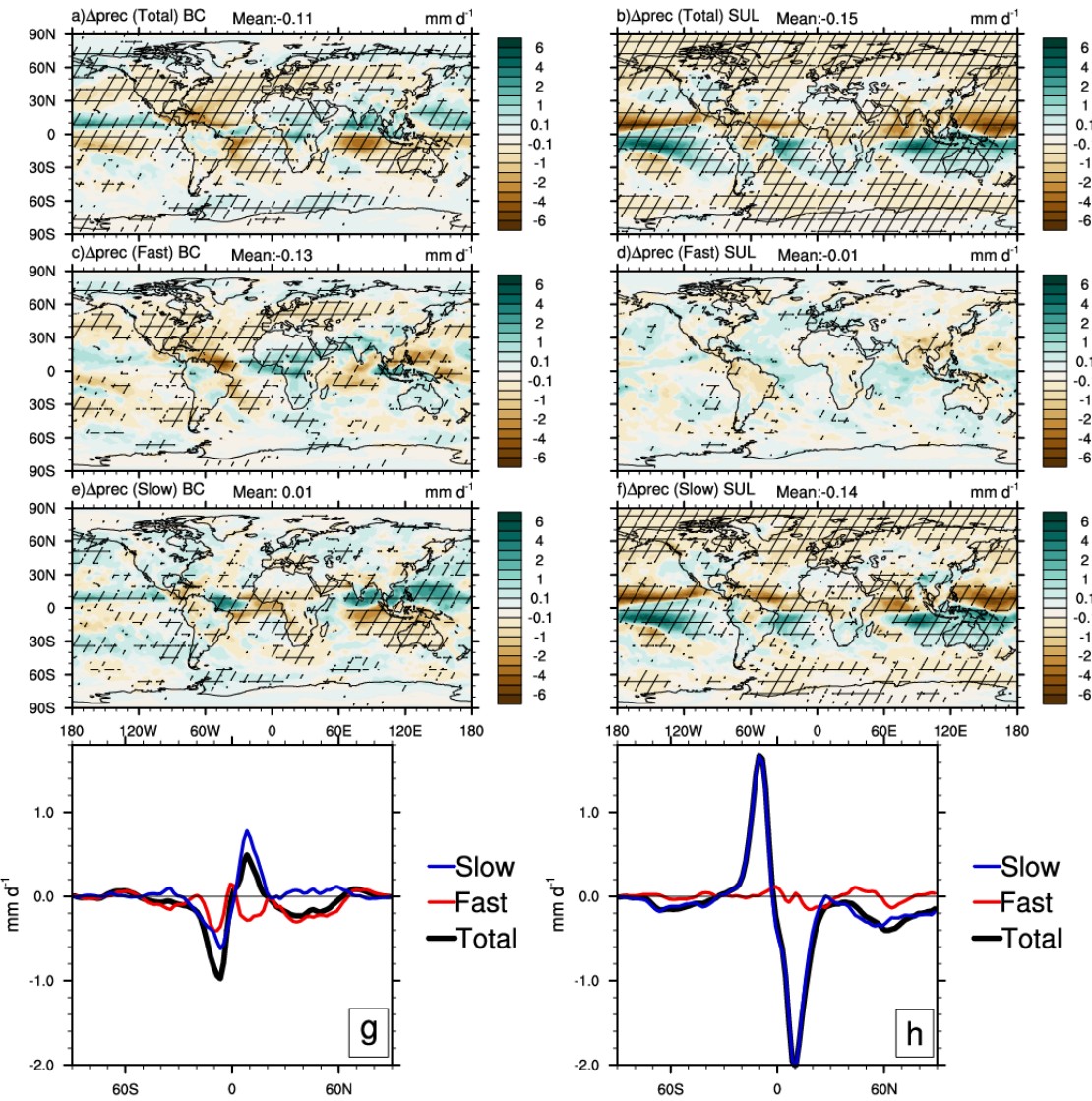


**Figure 1. ECHAM6-HAM2 simulated geographical patterns of multi-annual mean precipitation change in response to**
**increasing (left column) 10 times BC emissions and (right column) 5 times SUL emissions for (first row) total, (second**
**row) fast, and (third row) slow responses. Hatching indicates where the changes are significant (90% confidence).**
**(fourth row) Zonal averages of changes in precipitation in terms of total, fast and slow responses to increasing (g) 10**
**times BC emission and (h) 5 times SUL emission.**










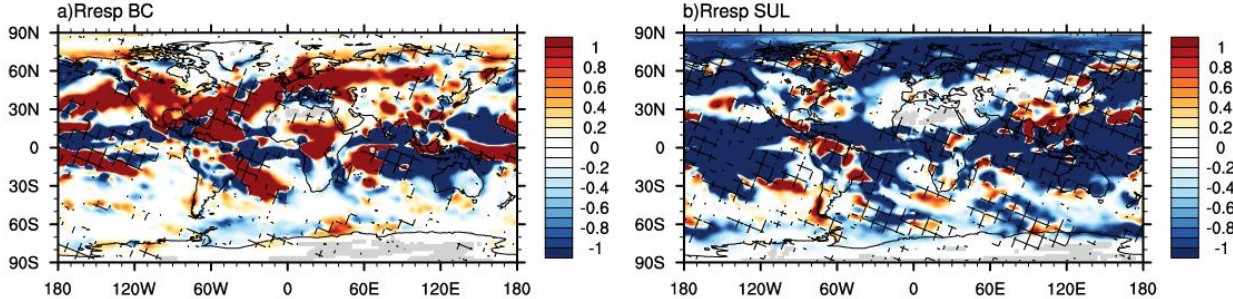



**Figure 2.** Response ratio of fast and slow responses ($R_{resp}$) (red denotes fast responses dominates the total responses
and blue indicate slow responses dominates) of fast and slow responses for (a) BC cases and (b) SUL cases. Results have
been normalised by total responses of precipitation. Hatching indicates the signs of fast and slow responses are same.
If $R_{resp}$ is around 0, contributions from fast and slow responses are similar. If $R_{resp}$ is larger than 0, the
total response is dominated by fast responses. If $R_{resp}$ is less than 0, the total response is dominated by slow
responses.



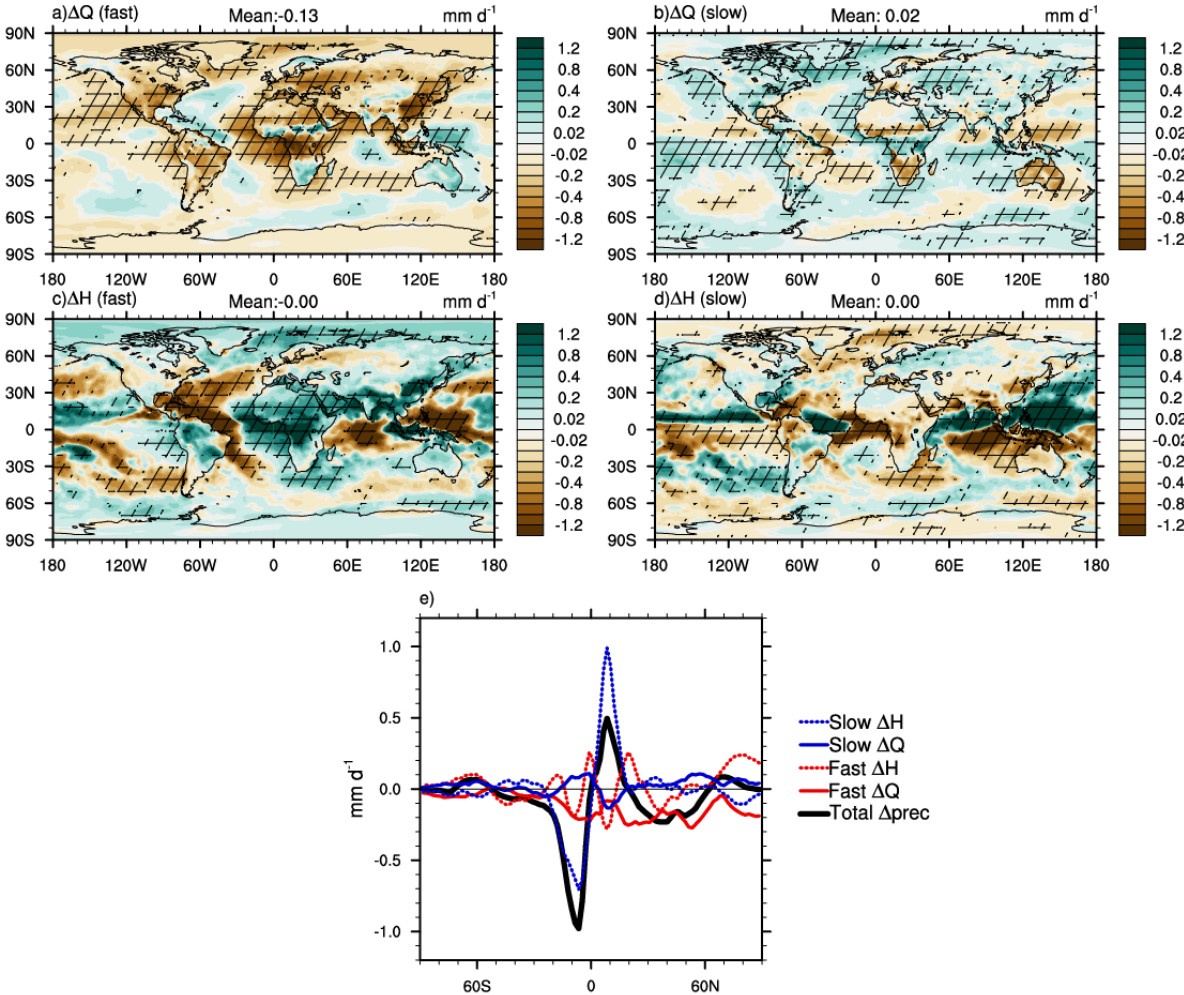

**Figure 3. ECHAM6-HAM2 simulated geographical patterns of multi-annual mean changes in (first row) atmospheric**
**diabatic cooling ($\Delta Q$) and (second row) dry static energy flux divergence ($\Delta H$) for (left column) fast responses and**
**(right column) slow responses to 10 times BC emission. Hatching indicates where the changes are significant (90%**
**confidence interval through bootstrapping methods). (e) The zonal mean of total precipitation response and its**
**decompositions, including fast and slow responses of diabatic cooling and dry static energy flux divergence. All of them**
**are shown in equivalent precipitation units of mm d$^{-1}$.**


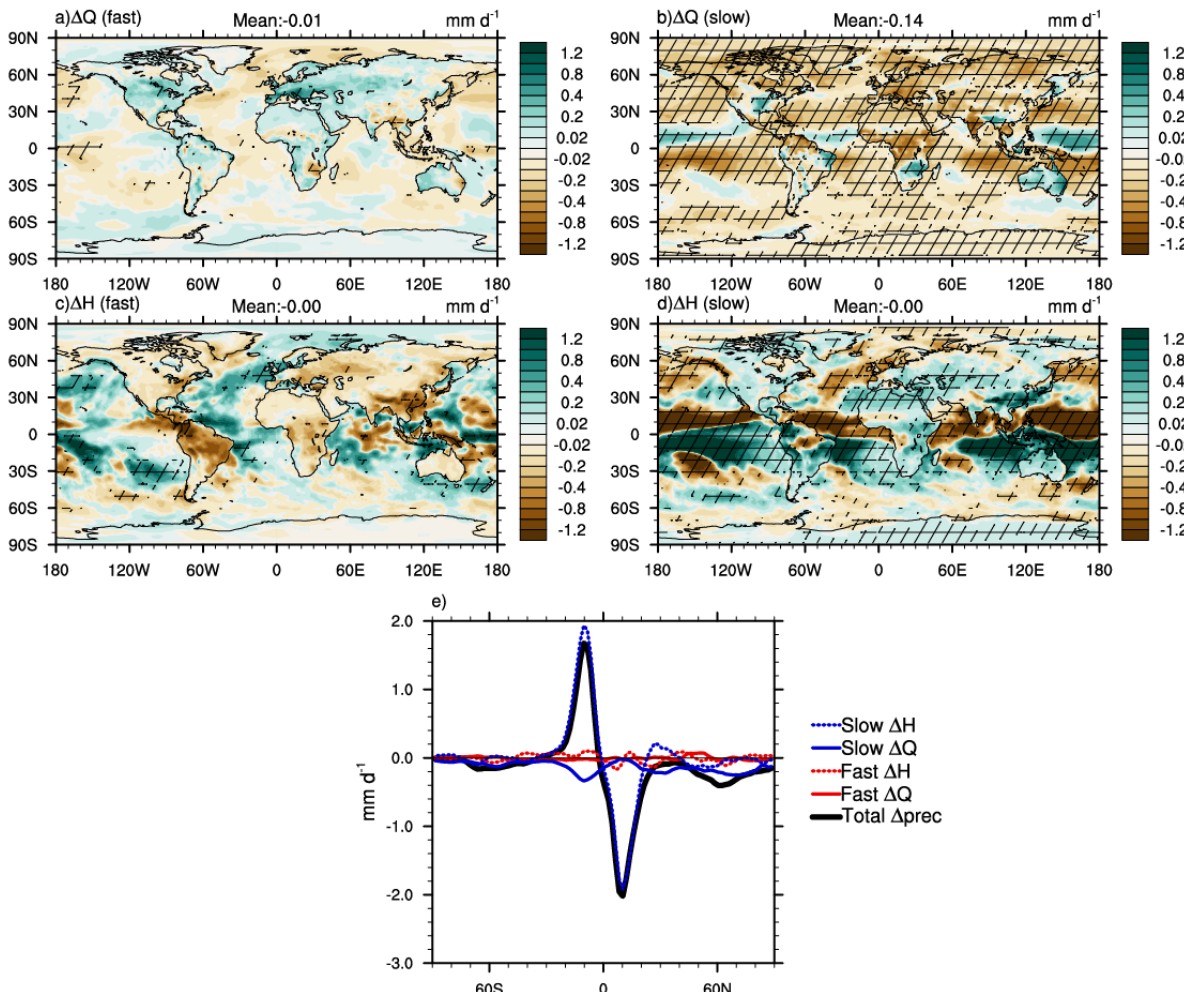

**Figure 4. ECHAM6-HAM2 simulated geographical patterns of multi-annual mean changes in (first row) atmospheric**
**diabatic cooling and (second row) dry static energy flux divergence for (left column) fast responses and (right column)**
**slow responses to 5 times SUL emission. Hatching indicates where the changes are significant (90% confidence interval**
**through bootstrapping methods). (e) The zonal mean of total precipitation response and its decompositions, including**
**fast and slow responses of diabatic cooling and dry static energy flux divergence. All of them are shown in equivalent**
**precipitation units of mm d⁻¹.**


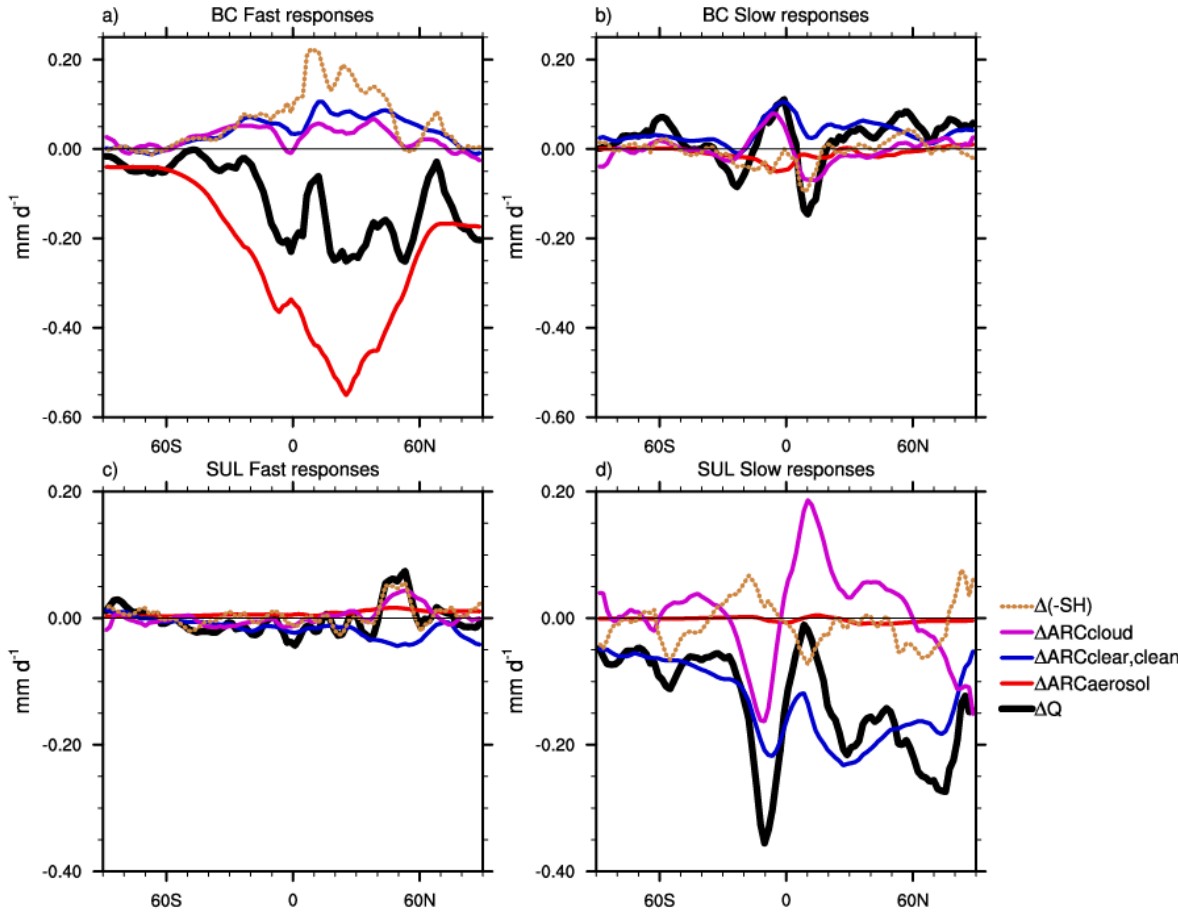

**Figure 5. ECHAM6-HAM2 simulated multi-annual zonal mean of decomposed changes in atmospheric diabatic cooling**
**($\Delta Q$), including ARC changes from aerosols ($\Delta \mathrm{ARC}_{aerosol}$), clouds ($\Delta \mathrm{ARC}_{cloud}$), clear-clean sky ($\Delta \mathrm{ARC}_{clear,clean}$),**
**downward sensible heat flux($\Delta(-SH)$) for (a) fast responses in the BC case, (b) slow responses in the BC case, (c) fast**
**responses in the SUL case, and (d) slow responses in the SUL case. All items are shown in equivalent precipitation units**
**of mm d$^{-1}$.**

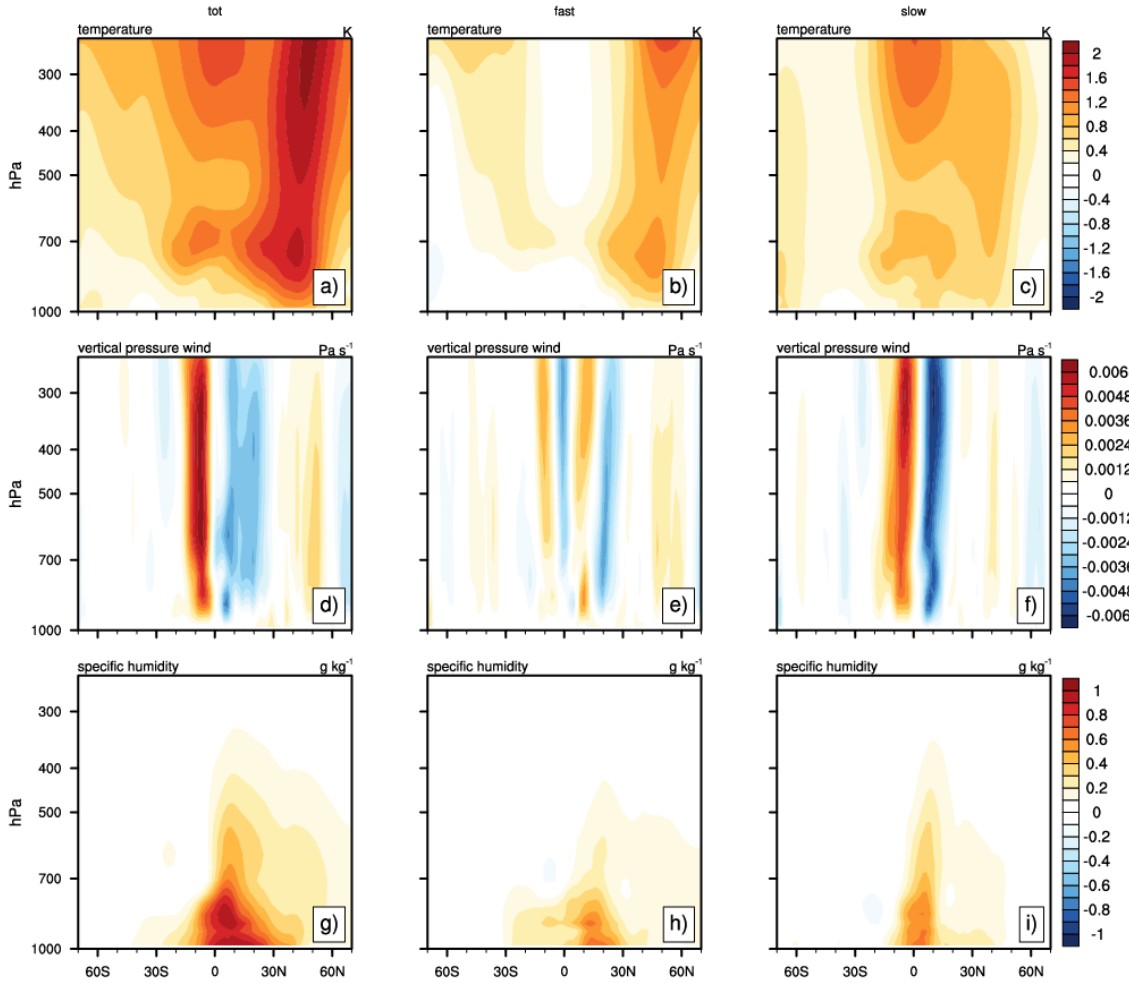

**Figure 6. ECHAM6-HAM2 simulated multi-annual (left column) total, (middle column) fast, and (right column) slow responses of zonally averaged (a, b, c) temperature, (d, e, f) vertical pressure velocity, and (g, h, i) specific humidity in response to 10 times BC emission. Blue colours indicate large-scale ascent, and red colours indicate large-scale descent in d-f.**


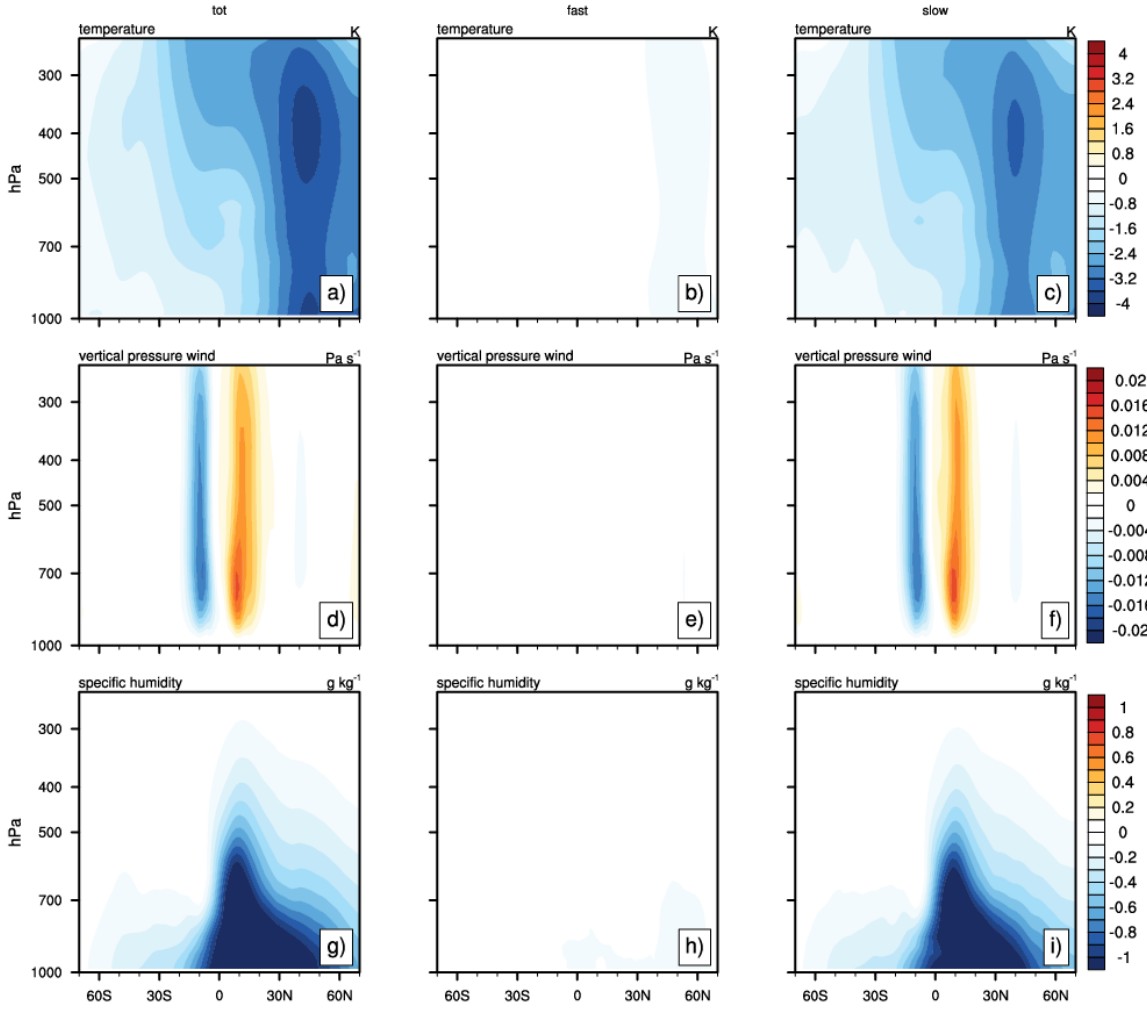


**Figure 7. ECHAM6-HAM2 simulated multi-annual (left column) total, (middle column) fast, and (right column) slow responses of zonally averaged (a, b, c) temperature, (d, e, f) vertical pressure velocity, and (g, h, i) specific humidity in response to 5 times SUL emission. Blue colour indicates large-scale ascent, and red colour indicates large-scale descent in d-f.**

931

932
933