# Peer review of "On the Contribution of Fast and Slow Responses to Precipitation Changes Caused by Aerosol Perturbations"

_Atmospheric Chemistry and Physics, 2020_

## Author Comment (AC1)

**We would like to appreciate the helpful comments from the reviewers, which have allowed us to improve the manuscript. The following are our responses to the reviewer comments, with the reviewer comments in italic and our response in bold.**

*Anonymous Referee #1*

*The manuscript submitted by Zhang et al. uses the global Earth system model ECHAM6-HAM to investigate how idealized perturbations of BC and SO4 influence precipitation. They divide precipitation responses into the rapid and the slow surface-temperature-mediated response, and find – as earlier work – that BC-induced precipitation changes are dominated by the rapid response while SO4-induced changes are driven mostly by the slow response. Authors analyze both rapid and slow responses in terms of the energy budget, and further decompose every term into contributions from clouds, aerosols, and clean-clear-sky. This approach allows for a thorough understanding of which processes underly the different changes. The paper is very well written and based on sound scientific analyses. While the results are unsurprising, I believe that the clear scope, the thorough method, and the systematic and accurate account of results makes the paper a good contribution to the field, and I recommend the paper to be published after relatively minor revisions.*

*General comment:*

*As the manuscript is so well worked through and so well written my comments are few.*

*Specific comments:*

- *L 79: "lacks agreement in both model simulations and observations". Please clarify – is there little agreement between typical model results and what observations show, or is there both deviations between models and observations as well as large inter-model disagreement?*

**Sorry that the previous statement was unclear. This sentence now has been rephrased for clarity: *"It can therefore alter cloud microphysics and subsequent precipitation formation. However, the susceptibility of precipitation to sulphate aerosols (and the precursors) shows large discrepancies in satellite-estimated precipitation susceptibility to aerosols from several products (Bai et al., 2018; Haynes et al., 2009), and a broad inter-model spread (uncertainty) in GCMs (Ghan et al., 2016; Samset et al., 2016). Some studies also found that the sensitivity of precipitation to sulphate aerosols differs between model-simulated and satellite-estimated results, in terms of magnitude and sometimes in sign (Liu et al., 2020; Wang et al., 2012)."***

- *L 88: "Fast and slow responses are essential in determining changes in precipitation" This one is a bit unclear as to what the authors mean. Do you perhaps mean that identifying fast and slow responses is essential to determining the causes or processes behind the precipitation change? Consider rewording this sentence.*

**We have rephrased this sentence: *"Distinguishing contributions from fast and slow responses are essential for understanding the mechanisms that cause the precipitation changes."***

- *L 128: "on different scales" – Please clarify, is it different timescales (fast vs slow) you mean here?*

**Since we are examining fast and slow responses (timescales) and precipitation changes on both global and regional (spatial) scales, it was supposed to be different temporal and spatial scales. We have clarified it now in the manuscript: *"What is the dominant energetic term in***

*precipitation responses to absorbing/non-absorbing aerosol perturbations on different spatial and temporal scales?"*

- *L 198: I'm not familiar with ECHAM6-HAM – does this model include any microphysical effects of BC, allowing BC to function either as a condensation or as an ice nuclei?*

**In ECHAM6-HAM2 (the version used in this study), BC can be activated as a cloud condensation (through internal mixing) or ice nuclei based on its size and hygroscopicity. The aerosol module HAM2 (Stier et al., 2005; Tegen et al., 2019; Zhang et al., 2012) predicts the mass and number concentrations of aerosols based on their sizes. HAM2 describes five species of aerosols (including BC) into 7 lognormal modes (M7) based on their sizes (nucleation, Aitken, accumulation, and coarse modes) and hygroscopicity (soluble and insoluble) (Vignati et al., 2004). The activation of CCN to cloud droplets is adopted from Abdul-Razzak and Ghan (2000), which is based on Köhler theory (Köhler, 1936). Freshly emitted BC is assumed hydrophobic and does not act as cloud condensation nuclei. However, subsequent condensation of sulfuric acid and mixing with hydrophilic sulphate aerosols will increase its hygroscopicity so that internally mixed BC particles can activate as CCN (Stier et al., 2006).**

**In HAM2.3, BC can act as ice nuclei through heterogeneous freezing, but only in the accumulation and coarse mode (Neubauer et al., 2019).**

**We now have added descriptions in the manuscript:** *"It should be noted that freshly emitted BC is assumed hydrophobic and does not act as cloud condensation nuclei. However, subsequent condensation of sulfuric acid and mixing with hydrophilic sulphate aerosols will increase its hygroscopicity so that internally mixed BC particles can activate as CCN (Stier et al., 2006). In HAM2.3, BC can act as ice nuclei through heterogeneous freezing, but only in the accumulation and coarse mode (Neubauer et al., 2019)."*

- *L 285: Please fix the subscript in "Rresp"*

**Fixed.**

- *L 311: Just for easier reading: please add "fast" before "precipitation" in the beginning of this line*

**Thanks. Done.**

- *L 326: What causes the slow BC-induced change in atmospheric cooling?*

**The increased slow-response atmospheric cooling in BC case is generally caused by a warmer atmospheric column. Increased BC emission leads to an increase in surface temperature in slow responses (Table 1), which eventually leads a warmer atmospheric column (Figure 6c). From an energetic perspective, this atmospheric cooling is mainly contributed by clear-clean sky LW cooling ($ARC_{clear,clean}$) (Table 1 and Figure 5b), due to the increased atmospheric column temperature (Planck feedback). We added some descriptions after this sentence**: *"This increased precipitation in the slow-response is caused by the associated increase global temperature (Figure 6c) (Table 1). From an energetic perspective, it is mainly associated with the clear-clean sky LW cooling ($ARC_{clear,clean}$) (Table 1 and Figure 5b) as a result of increased atmospheric column temperature (Planck feedback)."*.

- *L 343: Again, just for easier reading it would be good to remind the reader that this sentence is regarding SO4, not BC (which is mentioned on the previous line).*

**Fixed.**

- *L 490: "largely balance and less balanced" – does this refer to global and extratropical respectively? Please consider rewording the sentence to make this more clear.*

**Sorry that our previous statement was unclear. This sentence now has been rephrased: *"Generally, changes in ARC and latent heat from precipitation are largely balanced on global scales and in the extra-tropics (Dagan et al., 2019b). However, these two terms are less balanced in the tropics due to efficient local energy transport."***

- *FIG 2: Consider adding a short sentence in the caption explaining what values of e.g. -1, 1 and 0 would mean, to make the figure easier to understand without reading the manuscript text.*

**Done. Statement has been added: "If $R_{resp}$ is around 0, contributions from fast and slow responses are similar. If $R_{resp}$ is larger than 0, the total response is dominated by fast responses. If $R_{resp}$ is less than 0, the total response is dominated by slow responses."**

- *FIG 3 and 4: Please increase the font size of the legend in panels e)*

**Done.**

- *FIG 5: In the legend, you use e.g. ARCaero and ARCcc, while in the manuscript text and also in the figure caption it differs whether you use ARCaerosol /ARCaero or ARCcc/ARCclean,clear. Please make this consistent throughout the text and figures. Also, please increase the font size of this legend.*

**Fixed accordingly.**

- *FIG 6 and 7: Could you, to help the reader, consider adding a short sentence in the caption, explaining that red/blue is descent/ascent in d-f?*

**Good point. We have added the description in the captions accordingly: "Blue colours indicate large-scale ascent, and red colours indicate large-scale descent in d-f."**

*Anonymous Referee #2*

*The global aerosol-climate model ECHAM6-HAM is used to investigate the response of precipitation to perturbations of black carbon and sulfate based on examining changes in atmospheric energy budget on global and regional scales. Precipitation responses are divided into the fast response due to black carbon and the slow response due to sulfate. Overall, the manuscript is well written and good analyses have been provided. However, there is an important concern regarding the approach of the study, which includes application of a general circulation model (GCM) for unraveling precipitation responses on regional scales to aerosol perturbations. Further comments are listed below.*

**Thanks for the suggestions. As for the point of relatively low resolution used in this study, please see our detailed response to the third comment below.**

*Although the manuscript is generally well written, some minor syntax errors need to be fixed. In addition, long sentences should be avoided. For example, in line 60-62.*

**We have proofread again to avoid long sentences in the manuscript and fixed the syntax errors. For example, sentences in lines 60-62 now have been broken into shorter sentences:** ***"However, in the tropics, horizontal gradients of dry static energy are small due to the weak Coriolis force. Therefore, local strong diabatic heating perturbations can lead to thermally direct circulations that drive convergence/divergence of moisture and dry static energy. This low-level convergence of mass and moisture can lead to vertical motions and thus an increase in precipitation. So rainfall does not necessarily have to positively correlate with diabatic cooling (Dagan et al., 2019)."***

*Lines 31-38. The uncertainty of aerosol indirect effects or, maybe more precisely, different or even opposite conclusions, which have been obtained for the indirect effects of aerosols in different studies, are related to different environmental conditions, including relative humidity, vertical wind shear and cloud types. This needs to be clarified in this paragraph and the works of Khain et al. (2008), Khain (2009) and Alizadeh-Choobari (2018) can be cited.*

**We have now added more illustrations on the uncertainty of aerosol indirect effects at the end of this paragraph:** ***"For example, satellite-estimated and model-simulated aerosol-cloud interactions show large discrepancies in terms of magnitude and even in sign (e.g. Ackerman et al., 2004; Rosenfeld et al., 2019; Wang et al., 2012). Disagreements between different studies can be attributed to methodologies (Gryspeerdt et al., 2014), model uncertainties (White et al., 2017) and, importantly, are often related to differences in environmental conditions, such as relative humidity, dynamic background, cloud types, stability (Alizadeh-Choobari, 2018; Khain, 2009; Khain et al., 2008; Lohmann et al., 2007; Zhang et al., 2016)."***

*Line 76. The recent related work of Keshtgar et al. (2020) can be also cited here.*

**Done.**

*Line 105-108. Here the lack of regional studies is emphasized to convince conduction of the study. However, a GCM) with a relatively low resolution (1.9 degree) is applied in this study, which is not appropriate to demonstrate precipitation responses on regional scales. Considering the fact that precipitation highly varies on regional scales, the authors should explain how*

*application of a GCM can be helpful to unravel responses of precipitation on regional scales to aerosol perturbations.*

**We agree that high resolutions are desirable for the analysis of regional precipitation changes. However, climate models in such configurations have been widely used in this context (the entire CMIP and PDRMIP exercises rely on this) and been shown to have skills in examining regional rainfalls as well as their responses (e.g. Liu et al., 2018; Myhre et al., 2017; Samset et al., 2016). Increasing resolution while retaining parameterised convection, as done in many regional climate modelling studies, raises other concerns as many assumptions underlying these parameterisations are no longer valid (Prein et al., 2015). Ultimately, such work should be conducted in cloud resolving configurations (which would also allow to couple aerosols directly to the convection, an effect that is not currently represented) and work is ongoing to develop the required tools. However, it will still be decades before these are routinely available. In the context of the focus of this work, with focus on constraints from the energy budget and the underlying physical constraints in general, GCMs are in fact a very robust tool (and ECHAM6-HAM is, unlike other GCMs or many cloud resolving models, fully energy conserving). We therefore believe our approach to be robust, in-line with a vast body of literature on this very topic (e.g. Jordan et al., 2018; Myhre et al., 2017; Roeckner et al., 2006; Samset et al., 2016; Shawki et al., 2018; Samset et al., 2016).**

**We should point out that internal variability on regional scales is significant, in particular in coupled simulations. However, since we are examining the average of last fifty years results instead of the transient evolution, the impacts from internal variability should be small in this case. Therefore, this does not take away from our analysis of physical constraints on precipitation changes. Nevertheless, we have pointed out the associated uncertainties in the manuscript.**

*Line 153. It should be explained why BC and Sulfur dioxide have been increased differently, and how it can impact the obtained results.*

**This work focuses on fundamental physical constraints in semi-idealised setups. The scaling factors of ten times BC and five times SO2 were chosen following the PDRMIP protocol (Samset et al., 2016) with the aim to increase signal to noise for manageable integration periods. Different scaling factors for different species reflect their different radiative effects (Myhre et al., 2017).**

**Considering the aerosol burdens are different between BC and SUL cases (Figure S1), in the manuscript, we have normalised the results in Figure 2. As for other figures, we have avoided directly comparing the values between BC and SUL cases and focusing on the different mechanisms behind them (e.g., comparing the contributions from fast and slow responses). We have added some statements in the manuscript accordingly*: "We chose the multipliers of aerosol emissions differently here is to make the aerosol effects statistically large enough and keep their radiative forcing at the same magnitude (Myhre et al., 2017). Another reason is to make our results comparable with PDRMIP work (Samset et al., 2016)."***

*Anonymous Referee #3*

*General comments*

*This study evaluated the impact of artificially perturbed aerosol emissions on precipitation in ECHAM-HAM and investigated the underlying mechanisms by looking at individual energetic terms. Following an earlier study, the authors separated the fast and slow responses using atmosphere-only and slab ocean simulations. The major conclusion on global mean precipitation response is similar to previous studies, i.e. both BC and sulphate emission increases reduce the precipitation, but the impacts are through different processes (fast process for BC, but slow process for sulphate). They further compared the decomposed terms of atmosphere radiative cooling and found the largest terms are from aerosol radiative effect for BC and clear-clean sky effect (mainly due to reduced water vapor) for sulphate. Overall I think this study is useful and it provides new information to understand the climate impact of aerosols. Nevertheless, I think the description of slab ocean simulations should be improved and the interpretation of the results needs some clarifications. Below please find my specific comments.*

**Thanks for the comments. We have added more descriptions to introduce the slab ocean simulations in the manuscript. More descriptions have been added to improve the manuscript for clarifications. Please also see our responses to the specific comments below.**

*Specific comments*

*Page 1, abstract, line 24: Non-absorbing aerosols can decrease surface temperature by scattering shortwave radiation, which is a fast process. Do you mean a different mechanism?*

**Non-absorbing aerosols scatter shortwave back to space, which is a fast process and causing negative radiative effects at both TOA and the surface. However, the decrease of global-mean surface temperature is a slow process, due to the high heat capacity of oceans. Nevertheless, the land surface temperature change is still a fast process. Related statements can be also found at Section 3.1: "As SUL decreases SW radiation reaching the surface, the global-mean temperature decreases around 2K on a relatively long timescale due to the high capacity of oceans (a slow process)."**

*Page 3, line 84-85: do you mean the atmosphere-only model?*

**Yes, it is supposed to mean the atmosphere-only models here. We have rephrased it to make it clear. Now it states: "It should be noted that even though SST is unchanged in atmosphere-only models, the land surface temperature is generally still allowed to vary (Stjern et al., 2017)."**

*Page 3, line 102: "which is" -> "which are"*

**Fixed.**

*Page 3, line 104: is LW radiative cooling the only term? How about SW changes due to aerosol-cloud interactions?*

We noted that aerosols could significantly reduce downward SW radiation through aerosols-cloud interactions. However, this cooling is different from atmospheric radiative cooling, which is the difference of radiative fluxes between TOA and the surface. Despite the significant negative radiative forcing at TOA, non-absorbing aerosols do not significantly modify atmospheric radiative absorption, as they act to decrease net SW radiative fluxes at both the surface and TOA in the same way. Therefore, most of the radiative cooling is from LW radiation from clouds rather than SW radiation. We have shown that its sign and magnitude depend on the temperature (height) at both cloud top and bottom as well as on the ice concentration at cloud top (see Figure S2 for baseline $ARC_{cloud}$). Related statements can also be found in Section 2 when introducing each decomposed ARC term: *"It is worth noting that $\Delta ARC\_aerosol$ only includes direct interactions with radiation here and is much more sensitive to absorbing aerosol burden rather than non-absorbing aerosols. Despite the significant negative radiative forcing at TOA (Boucher et al., 2013), non-absorbing aerosols rarely modify atmospheric radiative absorption, as they act to decrease net SW radiative fluxes at both the surface and TOA through scattering solar radiation. Non-absorbing aerosols can affect atmospheric radiative absorption via changing absorbing aerosol life cycles (Stier et al., 2006), but the impacts can be very small. It should also be noted here that changes in ARC_cloud include aerosol indirect effects (interactions with clouds) on ARC and cloud feedbacks in slow responses, but most of the changes are from LW radiation from clouds (e.g., Lubin and Vogelman, 2006) rather than SW radiation. And its magnitude depends on the temperature (height) at both cloud top and bottom high as well as ice concentration at cloud top (see Figure S2 for baseline ARC_cloud). As aerosol effects on convective clouds are not explicitly simulated in ECHAM6-HAM2 (or most GCMs) yet, changes of ARC_cloud from convective clouds are mostly caused by aerosol-induced changes in dynamics. Baseline $\Delta ARC\_aerosol$, $\Delta ARC\_cloud$, and $\Delta ARC\_clear,clean$ can be seen in supplementary file (Figure S2, S3, S4)."*

*Page 5, line 153: is the emission perturbation for all or anthropogenic sources only?*

For SO2 emission, this work only increases the anthropogenic sources, but for BC emissions, the increase includes both anthropogenic and some natural sources. As stated in the earlier part of Section 2, biomass burning emissions are also from ACCMIP dataset, including both natural and anthropogenic biomass burning (Lamarque et al., 2010). This is because the emissions include both anthropogenic and naturally occurring wild fires. The anthropogenic contribution to wildfire emissions is assumed to dominate but subject of significant uncertainties (e.g. Lamarque et al., 2010; van Marle et al., 2017). However, the purpose of the study is to understand the mechanisms of absorbing and scattering aerosols on precipitation, which his source independent. For this work it is therefore less important (but still of interest) to separate anthropogenic and natural contribution. We have added related descriptions in the Method part: *"It should also be noted that the increase of BC emissions here includes both anthropogenic and natural sources. This is because the biomass burning emission, as a large source of BC, includes both anthropogenic and naturally occurring wild fire emissions. The anthropogenic contribution to wildfire emissions is assumed to dominate but is subject to significant uncertainties (e.g. Lamarque et al., 2010; van Marle et al., 2017). However, the increases in SO2 emissions are all anthropogenic, because the sources of volcanic and sulphur are kept the same. The main purpose of this work is to better understand the mechanisms of aerosol-precipitation interactions, with a focus on, but not limited to, anthropogenic aerosol effects."*

*Page 5, line 158: please provide more information on how the MLO model was setup in your simulations. For example, how were the QFLUX data derived for MLO?*

The mixed layer ocean is described as 50 meters in depth (Dallafior et al., 2016). The ocean heat transport term (also known as the Q flux) is prescribed (calculated from the 30 year PDRMIP baseline present day setup) (Myhre et al., 2017), which also means the ocean dynamics are unchanged. Therefore, the changes in sea surface temperature are caused by local responses to net surface heat flux, and the effects from changes in ocean circulations are omitted. We have added descriptions accordingly: *"We run the simulations for 100 years with a mixed layer ocean (MLO), which is described as 50 meters in depth (Dallafior et al., 2016). The ocean heat transport term (also known as the Q flux) is prescribed, which also means the ocean dynamics are unchanged. Therefore, the changes in SST are caused by local responses to net surface heat flux, and the responses in ocean circulations are omitted."*

*Page 5, line 160: the time when a slab ocean model reaches equilibrium state is dependent on the model physics and how large the model response (to the perturbation) is. Different models might need different time to reach equilibrium. It's better to check it in your model (e.g. following the method used in Samset al. 2016).*

We acknowledge that it might take more than 100 years for a slab ocean model to fully equilibrate. Therefore in the manuscript, we claimed it is an approximate equilibrium, as in Samset et al., 2016. We also performed a Gregory-style regression (Gregory and Webb, 2008) here to check the equilibrium for the BC and SUL cases respectively. It can be seen that for the BC experiment, the net energy flux at TOA is rapidly reduced to lower than 0.1 W/m2 within ten years after the increase of BC emission, and it is likely to reach equilibrium after 50 years. For the SUL case, the energy imbalance is significantly reduced and reaches a near-equilibrium after 50 years run as well, but it is suggested that more than 100 years simulation is needed to fully equilibrate. So the total responses of surface temperature to 5 times SUL should be even lower (more negative). Considering the purpose of our study is to understand the mechanisms of precipitation responses to aerosols, the exact equilibrium is not critical here and our conclusions still apply to an approximate equilibrium. Nevertheless, we have added some statements about the approximate equilibrium in Section 2, and added this Gregory-style regression plot in the supplementary file.

[Figure]

**Figure. A Gregory-style (Gregory and Webb, 2008) regression between changes in surface temperature and changes in net energy flux at the top of the atmosphere, for (a)10 times black carbon and (b) 5 times sulphur dioxide perturbation experiments, respectively. Each dot**

**represents the annual average of each individual year (100 years in total). The solid dots represent the first 8 years since the aerosol perturbations were added.**

*Page 5, line 170: it would be useful for the readers learn more details on how the energy transport term is diagnosed, e.g. did you calculate it online or offline?*

**It is calculated offline, as a residual by using the energy budget equation. We have added some descriptions accordingly:** *"H is calculated offline, as a residual by using the energy budget equation."*

*Page 6, section 3.1, table 1: please check the numbers in the table. It seems to me that L&P is not equal to &ARC - &SH for most cases. Also, it's better to use same unit for the terms shown in the table and in the figure. Right now one (table) in W/m2, and the other (figure) in mm/d.*

**Thank you very much for spotting this and apologies as there was a code issue for generation of this table. When calculating the global-mean responses of precipitation and surface temperature, which only sampled the 5th to 15th years (which was supposed to be the last ten years) for fast responses, and 50th to 60th year for total responses (which was supposed to be last fifty years). But the other terms are sampled over the correct periods. So the energetic terms did not add up. We have updated the numbers and converted the units into mm d-1 to keep consistent with other figures.**

*Page 11, line 401-402: based on the discussion on the fast/slow response, the reduced surface temperature is mainly due to the slow response. The sentence here reads like the dimming effect of aerosols (fast process) leads to reduced surface temperature and the major cooling in NH shown Figure 7a. Is this really what the author meant?*

**It is supposed to mean that the dimming effect from sulphate aerosols, which is a fast process, caused negative forcing at the surface. However, the decease of global surface temperature is controlled by ocean heat capacity and a slow process, in response to this negative forcing, and contributed to most of the cooling in NH shown. Figure 7a shows the total responses of atmospheric temperature to increased SUL emission, which is mainly contributed by the slow response (Figure 7c). We have rephrased the description in the manuscript:** *"
[revised manuscript text omitted]